# Mechanical Stimulation Alters Chronic Ethanol-Induced Changes to VTA GABA Neurons, NAc DA Release and Measures of Withdrawal

**DOI:** 10.3390/ijms232012630

**Published:** 2022-10-20

**Authors:** Kyle B. Bills, Dallin Z. Otteson, Gavin C. Jones, James N. Brundage, Emily K. Baldwin, Christina A. Small, Hee Young Kim, Jordan T. Yorgason, Jonathan D. Blotter, Scott C. Steffensen

**Affiliations:** 1Department of Biomedical Sciences, Noorda College of Osteopathic Medicine, Provo, UT 84606, USA; 2Department of Psychology/Neuroscience, Brigham Young University, Provo, UT 84602, USA; 3Department of Physiology, Yonsei University College of Medicine, Seoul 03722, Korea; 4Department of Cellular Biology and Physiology, Brigham Young University, Provo, UT 84602, USA; 5Department of Engineering, Brigham Young University, Provo, UT 84602, USA

**Keywords:** alcohol, physical medicine, dopamine, mechanoreceptors

## Abstract

Therapeutic activation of mechanoreceptors (MStim) in osteopathy, chiropractic and acupuncture has been in use for hundreds of years with a myriad of positive outcomes. It has been previously shown to modulate the firing rate of neurons in the ventral tegmental area (VTA) and dopamine (DA) release in the nucleus accumbens (NAc), an area of interest in alcohol-use disorder (AUD). In this study, we examined the effects of MStim on VTA GABA neuron firing rate, DA release in the NAc, and behavior during withdrawal from chronic EtOH exposure in a rat model. We demonstrate that concurrent administration of MStim and EtOH significantly reduced adaptations in VTA GABA neurons and DA release in response to a reinstatement dose of EtOH (2.5 g/kg). Behavioral indices of EtOH withdrawal (rearing, open-field crosses, tail stiffness, gait, and anxiety) were substantively ameliorated with concurrent application of MStim. Additionally, MStim significantly increased the overall frequency of ultrasonic vocalizations, suggesting an increased positive affective state.

## 1. Introduction

Alcohol use disorder (AUD) is a chronic relapsing disease that affects more Americans than all forms of cancer combined [1]. It leads to destructive psychological, physical, social, and economic consequences. It is estimated that over 28 million Americans are currently in need of treatment for alcohol abuse, resulting in over $249 billion in direct costs [2]. Making matters worse, only 13% of those needing intervention receive it. Further, despite the wonderful advances in our understanding of the neuropathophysiology of addiction, the success rate of treatment has not substantively changed over the last hundred years, with around 50% of those treated relapsing [1,3,4]. This represents approximately 6% of those suffering from the disease receiving effective treatment. A report ranking the different conditions relative to the “global burden of disease” found that ethanol ranked 3rd out of the 25 major contributors [5].

The current FDA-approved treatments for AUD are the sensitizing agent disulfiram, the mu-opioid receptor (MOR) antagonist naltrexone, and the neuromodulatory balancing agent acamprosate [6]. Many consider these pharmacological treatments to be woefully inadequate, and the public is begging for more efficacious treatments for AUD. Many are seeking alternative, non-invasive, non-pharmacological alternatives as personalized care and as adjuncts to self-help programs. Thus, more fundamental science and novel methods are needed to determine mechanisms underlying alternative approaches and improve outcomes.

The use of mechanoreceptor-based therapies (MStim) in the treatment of drug-abuse disorders is a largely unexplored field. Notably, several healthcare approaches are thought to have effects mediated in part by activation of mechanoreceptors, including osteopathic and chiropractic manipulation, acupuncture, and physical therapy, with compelling evidence of their potential to affect drug-seeking behaviors and treatment outcomes [7,8,9,10]. For example, electroacupuncture effects on cocaine psychomotor effects can be attenuated with ablation of the dorsal column/medial lemniscal pathway and appear to act through endogenous opioids [11,12]. Though these studies demonstrate anatomically site-specific effects, they are suggestive that activation of primary somatosensory fibers generally may attenuate the reinforcing effects of drugs of abuse.

As a mechanistic rationale, we have previously reported that 80 Hz activation of mechanoreceptors in the cervical spine, an area with a relatively high concentration of primary mechanoreceptors [13], inhibits ventral tegmental area (VTA) GABA neuron firing, enhances VTA dopamine (DA) neuron firing, and increases DA release in the nucleus accumbens (NAc) [14]. In the previous report, we demonstrated that the central effects disappear with superficial cutaneous stimulation or with higher frequency stimulation. Further, we have shown that VTA effects are driven by cholinergic interneurons (CINs) and delta opioid receptors (DORs) in the NAc [15]. These findings are salient considering the role that the mesolimbic circuitry plays in reward, dependence, and withdrawal. Midbrain DA neuron activity is involved in many aspects of reward seeking [16,17,18]. Although the prevailing dogma is that DA neurons mediate the rewarding and addictive properties of drugs of abuse [19], VTA GABA neurons have garnered much interest for their role in modulating DA release and perhaps as independent substrates mediating reward or aversion [20,21,22,23,24,25]. We have shown previously that acute administration of ethanol (EtOH), opioids, or cocaine inhibits VTA GABA neurons [20,21,22,23,26], leading to a net disinhibition of VTA DA neurons [27,28,29]. In contrast, during EtOH or opioid withdrawal, VTA GABA neurons become hyperactive [20,30] leading to decreased mesolimbic DA activity and release in the NAc [31,32,33,34,35]. This reduction in mesolimbic DA transmission is theorized to be the primary driver of relapse [36].

In this study, we investigate the role of targeted, platform-based MStim, as a simulation of mechanoreceptor-based treatment modalities. We utilized stimulation parameters similar to those we previously reported to be effective in modulating mesolimbic circuitry, as a potential treatment for EtOH-use disorder. To this end, we evaluate its ability to ameliorate chronic EtOH-induced changes to VTA GABA neuron firing, DA release in the NAc, behavioral indices related to withdrawal. We hypothesized that MStim would block chronic EtOH-induced desensitization of VTA GABA neurons and changes in NAc DA release in response to acute EtOH reinstatement and anxiety-related behaviors normally associated with chronic ethanol withdrawal (Figure 1).

## 2. Results

### 2.1. Amelioration of Chronic Ethanol-Induced Changes to VTA GABA Neurons by MStim

The effects of concurrent administration of 80 Hz MStim during chronic EtOH exposure were tested on VTA GABA neuron firing rate in the context of a reinstatement dose of EtOH (2.5 g/kg IP) (Figure 2). Baseline firing rate of GABA neurons in animals chronically exposed to EtOH was significantly higher (49.6 Hz) than that of animals that received concurrent MStim or saline with MStim (30.02 Hz and 35.94 Hz respectively; *F*_(2,11)_ = 5.1149, *p* = 0.0269; Figure 2D), consistent with the adaptations in VTA GABA neuron excitability with chronic EtOH in dependent animals that we have reported in mice and rats previously [20]. Mechanical stimulation produced altered GABA neuron response to reinstatement EtOH when compared to chronic EtOH and saline groups (*F*_(2,108)_ = 1348.799, *p* < 0.0001; Figure 2E). Administration of a 2.5 g/kg IP reinstatement dose of EtOH in chronically exposed animals produced a slight increase in average GABA neuron firing to 117.5 ± 0.039 %, *n* = 5) of baseline. The same injection in MStim treated animals caused a decrease in firing to 32.7 ± 0.043 %, *n* = 5) of baseline while in saline treated rats it produced a drop to 14.71% (±0.050, *n* = 5) of baseline. All groups were significantly different from each other (EtOH and MStim, *p* < 0.0001; EtOH and Sal, *p* < 0.0001; Sal and MStim, *p* < 0.0001; Figure 2F). between MStim and saline groups (*p* = 0.046). Thus, MStim, when given concurrently with chronic intermittent EtOH exposure to 2.5 g/kg EtOH, blocks chronic EtOH-induced desensitization of VTA GABA neurons to reinstatement exposure.

### 2.2. Amelioration of Chronic Ethanol-Induced Changes to Dopamine Release in the NAc by MStim

As VTA GABA neurons are critical regulators of VTA DA neural activity and DA release in the NAc, we evaluated the effects of MStim on DA release. To determine if blockage of chronic EtOH-induced changes to VTA GABA neuron firing rate by MStim translates to alterations in DA release in the NAc, we measured DA release from dialysate samples collected from an NAc cannula before and after a reinstatement (Figure 3). Groups tested included a group naïve to both MStim and EtOH, EtOH, EtOH + MStim and saline + MStim. Animals naïve to both EtOH and MStim exhibited a rise in DA levels from 20 min to 80 min post-injection and significant differences were noted at 20–80 min when compared to time point 0 (20 min, *F*_(3,13)_ = 6.0631, *p* = 0.0082; 40 min, *F*_(3,13)_ = 3.4890, *p* = 0.0471; 60 min, *F*_(3,13)_ = 3.8812, *p* = 0.0368; 80 min, *F*_(3,13)_ = 5.0333, *p* = 0.0157; Figure 3A). Significant differences arose when the EtOH group was compared to the EtOH + MStim and naïve groups at 20 min and 80 min post injection (20 min: EtOH vs. EtOH + MStim, *F*_(3,24)_ = 3.4964, *p* = 0.0310; 80 min: *F*_(3,23)_ = 9.0058, EtOH vs. EtOH +MStim, *p* = 0.0004, EtOH and naïve, *p* < 0.0001; Figure 3B,C). Interestingly, at 80 min post injection, significant differences arose between the Saline + MStim group (91.058 ± 8.34% baseline DA) and both EtOH + MStim (138.2 ± 5.93% baseline DA) and naïve (151.56 ± 10.20% baseline DA) groups (Figure 3C, *p* = 0.0234 and *p* = 0.0022 respectively), suggesting that MStim treatment alters DA release in response to EtOH in both previously exposed and naive animals.

### 2.3. Amelioration of EtOH Withdrawal Symptoms and Anxiety by MStim

To assess the behavioral relevance of MStim on indices of chronic EtOH withdrawal, rats in active withdrawal (24 h after last dose) were evaluated for open-field crosses, rearing behavior, tail stiffness and gait patterns (Figure 4). Over a 30 min period, rats were assessed for the number of times they engaged in rearing. Chronic EtOH exposure alone reduced rearing, an effect ameliorated by MStim treatment (*F*_(2,18)_ = 14.8194, *p* = 0.0002; Figure 4A). The chronic EtOH rats engaged in rearing on average 22.6 ± 4.51 times, which was significantly fewer times than rats treated with concurrent MStim (*n* = 8 in all groups; *p* = 0.0011 for EtOH + MStim; *p* = 0.0003 for EtOH and Sal). Mstim-treated rats reared 46.3 ± 3.29 times, while saline treated rats reared 49.9 ±3.65 times). Open-field crosses were defined as the number of times the rats crossed through the center 1/3 of the chamber in 30 min and were also assessed. Mechanical stimulation blunted the Chronic EtOH-induced reduction in the number of times rats engaged in open-field crosses (*F*_(2,18)_ = 18.8606, *p* < 0.0001; Figure 4B). The chronic EtOH rats crossed 14.1 ± 1.94 times). This was significantly fewer than the 29.9 ± 1.77) and 31.14 (±2.71) crosses engaged in by the MStim and saline groups, respectively, (*n* = 8 in all groups; *p* = 0.0002 for EtOH + MStim; *p* < 0.0001 for EtOH and Sal; Figure 4E–G). Tail stiffness was increased in the chronic EtOH when compared to the MStim or saline groups (*F*_(2,18)_ = 36.1579, *p* < 0.0001; Figure 4C). Tail stiffness in the chronic EtOH group was significantly different from the other two and was scored 3.57 ± 0.202 while the MStim and saline groups scored 1.86 ± 0.261 and 1.14 ± 0.143, respectively (*n* = 8 in all groups; *p* < 0.0001 for EtOH + MStim; *p* < 0.0001 for EtOH and Sal). Differences in gait patterns were noted between the three groups (*F*_(2,18)_ = 13.5789, *p* = 0.0003; Figure 4D). Gait scores for chronic EtOH animals were 3.43 ± 0.369). The MStim animals scored 1.71 ± 0.184) and the saline group scored 1.43 ±0.297. Chronic EtOH was significantly different from the other two (*n* = 8 in all groups; *p* = 0.0004 for EtOH + MStim; *p* = 0.0017 for EtOH and Sal).

Time in the open arm of the elevated-plus maze was significantly different between experimental groups (*F*_(3,20)_ = 4.6455, *p* = 0.0127; Figure 5). As expected, chronic ethanol withdrawal (141.41 ± 20.34 s) lowered time in the open arm when compared to air controls (210.17 ± 20.34 s; *p* = 0.0061). Importantly, MStim treatment, in rats chronically exposed to EtOH, increased time spent in the open arm (215.83 ± 12.50) when compared to rats exposed to chronic EtOH alone (141.41 ± 20.34 s; *p* = 0.0034). Thus, MStim, when applied concurrently with chronic intermittent EtOH, is sufficient to ameliorate certain behavioral indices associated with chronic EtOH withdrawal.

### 2.4. Effects of MStim on Ultrasonic Vocalizations

We have demonstrated previously that brief (60–120 s), low frequency (45–80 Hz), but not higher frequency (115 Hz), MStim from vibration motors implanted above the cervical vertebrae at the C7-T1 laminae enhanced both basal (178.4% peak increase at 60 min) and evoked DA release in the NAc (135.0% peak increase at 40 min) [15]. We hypothesized that MStim at 80 Hz might positively influence affective state due to its induction of DA release. To investigate, USVs were recorded during a 2.5 h session before and after MStim at 80 Hz. Calls were classified as being negative affective state-associated (20–30 kHz [37,38] or positive affective state-associated (35–90 kHz; Figure 6A). The average number of negative calls/session in a control 2.5-h session was 49.8 ± 16.9 (*n* = 5) and the number of positive calls was 201.2 ± 101.8 (*n* = 5). Compared to no vibration (Con), WBV at 80 Hz had no effect on the number of positive (*F*_(1,9)_ = 0.35, *p* = 0.57) or negative (*F*_(1,9)_ = 1.2, *p* = 0.30) calls or the ratio of positive/negative calls (*F*_(1,9)_ = 1.5, *p* = 0.26). The grand-averaged principal frequency of calls was less variable than the number of calls in Controls. MStim significantly increased the overall frequency of calls (Figure 6B; *F*_(1,21)_ = 15.1, *p* = 0.0009; mean control frequency = 37.7 ± 1.2 kHz; mean WBV frequency = 44.8 ± 1.4 kHz).

## 3. Discussion

We have previously reported that MStim acts on the NAc to increase local DA release and that this effect is mediated by activation of cholinergic and DORs. Further, projections from the NAc then cause a depression in VTA GABA neuron firing which results in VTA DA neuron disinhibition and a subsequent increase in firing. Because previous studies demonstrated that the effects are frequency and anatomically specific, an 80 Hz stimulation was selected and administered as described for this study. The current study was designed to investigate if these neuromodulatory changes are sufficient to alter chronic EtOH effects on VTA GABA neurons and withdrawal behavior. In the present study, the effects of 80 Hz MStim treatment, given twice daily for 15 min, concurrently with dependence-inducing chronic intermittent EtOH injections, were tested on various measures of EtOH withdrawal. While acute administration of EtOH reduces VTA GABA neuron firing, chronic intermittent EtOH exposure desensitizes VTA GABA neurons during reinstatement doses [20]. These effects are thought to occur through downregulation of D2 receptors in the VTA presumably due to EtOH-induced alterations in local VTA DA release [21]. Concurrent administration of MStim with chronic intermittent EtOH blocks these effects and changes VTA GABA neuron response to reinstatement from 117.5% of baseline to 32.7% of baseline. The blunting of the desensitization was not sufficient to return the GABA neuron response back to a naïve state as there were significant differences between the saline group and the EtOH + MStim groups. Mechanical stimulation has been previously shown to increase DA levels in the NAc for 2 h post-MStim [15]. These increased levels activate D1 and D2 expressing medium spiny neurons in the NAc that project back to the VTA and target non-dopaminergic neurons [39]. These projections could be responsible for MStim-induced changes in VTA GABA neurons response to chronic ethanol.

Dopamine levels were tested in the same chronic reinstatement paradigm. Ethanol naïve animals demonstrated a characteristic increase in DA levels [40] following ethanol injection. Further, animals treated with chronic EtOH alone did not exhibit the same increase in DA levels following EtOH administration. However, animals treated with MStim concurrently with chronic EtOH exposure exhibited DA release profiles significantly different than EtOH alone and nearly identical to those of a Naïve animal. This is suggestive that MStim alone is sufficient to alter mechanistic changes normally elicited by EtOH. It is noteworthy that saline with MStim also produced some changes in DA release akin to those elicited by chronic EtOH exposure. Chronic EtOH administration has been shown to increase expression levels of DORs while decreasing expression of mu opioid receptors [41]. As noted, we have previously shown that MStim produces increased translocation of DORs to cellular membranes, this commonality between EtOH and MStim could explain the desensitizing effects of MStim alone, without the context of chronic EtOH, of DA release over time.

We speculated that MStim amelioration of chronic EtOH effects on VTA GABA neurons, DA release, and behavioral indices of withdrawal might be due to its DA enhancing effects. Indeed, in our previous report, brief (2 min) 80 Hz vibration of cervical vertebrae significantly enhanced DA release for an hour [15]. It is conceivable that MStim enhancement of DA release is compensating for the reduction of DA release typically produced by chronic EtOH exposure, or perhaps acting like an acute dose of EtOH. We hypothesized that MStim would decrease negative and increase positive affect-associated USVs because of its effects on DA release, which are typically associated with reward. Ultrasonic vocalizations (USVs) are an ethologically valid index of negative and positive affective states in rodents [37]. The ascending mesolimbic cholinergic system initiates negative emotional states and the ascending DA system initiates positive emotional states [42]. The negative and positive states are signaled by 22 kHz and 50 kHz USVs, respectively. Although the precise interaction between calling behavior and motivational properties of drugs remains controversial, enhanced release of DA has been associated with decreased negative affect-associated USVs (22 kHz) and increased positive affect-associated USVs (50 kHz), which are initiated by the activation of DA receptors in the shell of the NAc [43]. We have recently reported that mechanical stimulation (MStim) at the C7-T1 laminae enhanced both basal and evoked DA release in the NAc [15]. We did not find any difference in the number of positive, negative, or ratio of positive/negative calls following MStim. However, we did find a marked increase in the frequency of vocalizations with MStim. In control sessions, the frequency of calls decreased, potentially due to habituation. In MStim sessions, the frequency of calls did not increase from baseline, but was significantly increased compared to controls over the 2.5 h session. These USV findings suggest an increased positive affect, perhaps associated with an enhancement in DA release, which we have shown recently develops over minutes to hours with vibration of the cervical vertebrae [15]. However, 22 kHz USVs have been associated with negative affective states and increase during alcohol and opioid withdrawal, suggesting that MStim is able to compensate for the negative affective state associated with EtOH withdrawal.

The MStim-induced alterations to chronic EtOH effects on VTA GABA neurons and DA release in the NAc ultimately manifest in blunting of the noted markers of dependence. This is extraordinary, as we have yet to discover any treatment, pharmacological or otherwise, that ameliorates alcohol dependence. The behavioral studies reported here buoy that finding as all measures of withdrawal gathered for this study were substantively improved by the addition of MStim to the chronic intermittent EtOH exposure paradigm employed to induce dependence. These findings are particularly relevant as they represent a non-invasive method of blocking chronic ethanol effects, including both physiological and affective. These data, when taken as a whole, provide a mechanistic rationale for future human studies to explore physical medicine modalities, including manipulative therapeutics and acupuncture as treatment options for the treatment of AUD. This study represents a mechanistic look at the effects of targeted mechanoreceptor activation in the context of EtOH exposure. It represents preliminary evidence that there is a mechanistic basis for future studies in humans.

## 4. Materials and Methods

### 4.1. Animals, EtOH Administration, and MStim Application

Male Wistar rats, weighing 250–320 g, from our breeding colony at Brigham Young University were used. Initial MStim testing with male and female rats demonstrated no significant difference in MStim-induced effects on mesolimbic neurons. Rats were housed in groups of 2–3 at a fixed temperature (21–23 °C) and humidity (55–65%) on a reverse light/dark cycle with ad libitum food and water. Rats were briefly anesthetized using isoflurane (4.0%) during injections to mitigate injection stress. Each received twice daily IP injections (0900 and 1700 h) of EtOH (2.5 g/kg; 16% *w*/*v* in saline) or saline for 14-days. Immediately following injections, rats were placed in a sound-attenuating chamber (57 cm × 57 cm × 50.8 cm), with floors constructed of 1 cm thick aluminum plate and isolated from the walls of the chamber with 4 vibration isolators at each corner. A low frequency effect (LFE) audio transducer (miniLFE ButtKicker, The Guitammer Company, Westerville, OH, USA) was suspended below the center of the floor. An 80 Hz, 500 mVpp sine wave was generated using a Wavetex Datron Universal Waveform Generator model 195 (San Diego, CA, USA) and amplified using a Crown model XLi 3500 (Los Angeles, CA, USA) amplifier. The vibration acceleration on the plate was 1.86 m/s^2^. All animals received a final injection at 1700 h and were tested at 0900 the following day. Experimental protocols were approved by the Brigham Young University Institutional Animal Care and Use Committee according to NIH guidelines.

### 4.2. Single Cell Electrophysiology

For recordings of VTA GABA neurons, a total of 15 rats were anesthetized using isoflurane and placed in a stereotaxic apparatus. Anesthesia was maintained at 1.5% with 2.0 L/min of air flow from a nebulizer driven by an oxygen concentrator. Body temperature was maintained at 37.4 ± 0.4 °C by a feedback regulated heating pad. With the skull exposed, a hole was drilled for placement of a 3.0 M KCl-filled glass borosilicate (BF150-86-10, Sutter Instruments, Novato, CA, USA) micropipette (2 to 4 MΩ; 1–2 µm inside diameter), driven into the VTA with a piezoelectric microdrive (EXFO Burleigh 8200 controller and Inchworm, Victor, NY, USA) based on stereotaxic coordinates [from bregma: 5.6 to 6.5 posterior (P), 0.5 to 1.0 lateral (L), 6.5 to 9.0 ventral (V)]. Potentials were amplified with an Axon Instruments Multiclamp 700A amplifier (Union City, CA, USA). Single-cell activity was filtered at 0.3 to 10 kHz (3 dB) with the Multiclamp 700A amplifier and displayed on Tektronix (Beaverton, OR, USA) digital oscilloscopes. Potentials were sampled at 20 kHz (12 bit resolution) with National Instruments (Austin, TX, USA) data acquisition boards in Macintosh computers (Apple Computer, Cupertino, CA, USA). Extracellularly recorded action potentials were discriminated with a World Precision Instruments WP-121 Spike Discriminator (Sarasota, FL, USA) and converted to computer-level pulses. Single-unit potentials, discriminated spikes, and stimulation events were captured by National Instruments NB-MIO-16 digital I/O and counter/timer data acquisition boards in Macintosh PowerPC computers. One cell was recorded and reported per rat to avoid confounds due to multiple doses of ethanol.

### 4.3. Characterization of VTA GABA Neurons

VTA GABA neurons were identified by previously-established stereotaxic coordinates and by spontaneous electrophysiological and pharmacological criteria [44]. VTA GABA neuron discharge activity characteristics included: relatively fast firing rate (>10 Hz), ON-OFF phasic non-bursting activity, and an initially negative spike with duration less than 200 µsec. GABA neurons were excited by iontophoretic DA (+40 nA) ejected from the recording pipette in some experiments. We evaluated only those spikes that had greater than 5:1 signal-to-noise ratio. After positive neuron identification, baseline firing rate was measured for 5 min to ensure stability.

### 4.4. Ultrasonic Vocalizations

Ultrasonic vocalizations (USVs) were recorded during a 2.5 h session in the same sound-attenuating boxes from a total of 10 rats. MStim (80 Hz) began 30 min into the USV recording and lasted for 15 min.

USVs were recorded using a 125 kHz ultrasonic microphone (miniMIC, Binary Acoustic Technology, Tucson, AZ, USA) attached to a PC computer running SCAN’R software (Binary Acoustic Technology). The recorded files were then analyzed for frequency, number and type of USV calls using DeepSqueak [43]. USV calls were separated into 15 min bins and the average principal frequencies for the control and treatment groups were compared using an analysis of variance (ANOVA). In a second analysis, calls were classified as positive affect-associated calls, negative affect-associated calls, or insignificant calls based on the principal frequency and duration of each call [45]. Calls between 20–30 kHz were classified as negative while calls between 35–90 kHz and between 10–150 ms in duration were classified as positive. Calls that did not fit these criteria were disregarded. The occurrence of positive and negative affect-associated calls in each bin was counted and a ratio of positive to negative calls for each bin and each rat was calculated. The average ratio for each time bin as grouped by treatment was then compared using an ANOVA. The average ratio of all of each rat’s calls were also compared using similar methods.

### 4.5. Microdialysis and High Performance Liquid Chromatography

Microdialysis probes (MD-2200, BASI) were stereotactically inserted into the NAc (+1.6 AP, +1.9 ML, −8.0 DV) of 7 rats per group. Artificial cerebrospinal fluid (aCSF; pH ~7.4 and osmolarity of 300–310 mOSm) composed of 150 mM NaCl, 3 mM KCl, 1.4 mM CaCl_2_, and 0.8 mM MgCl_2_ in 10 mM phosphate buffer was perfused through the probe at a rate of 3.0 µL/min. Samples were collected every 20 min for 4 h with reinstatement does of ethanol (2.5 mg/kg; IP) occurring after the first 2 h had elapsed. Determination of the DA concentration in microdialysis samples was performed using a HPLC pump (Ultimate 3000, Dionex, Sunnyvale, CA, USA) connected to an electrochemical detector (Coulochem III, ESA). The detector included a guard cell (5020, ESA) set at +270 mV, a screen electrode (5014B, ESA) set at −100 mV, and a detection electrode (5014B, ESA) set at +220 mV. Dopamine was separated using a C18 reverse phase column (HR-80, Thermo Fisher Scientific, Waltham, MA, USA). Mobile phase containing 75 mM H_2_NaO_4_P, 1.7 mM sodium octane sulfonate, 25 µM EDTA, 0.714 mM triethylamine, and 10% acetonitrile was pumped through the system at a flow rate of 0.5 mL/min.

### 4.6. Behavioral Measures of Withdrawal

Behavioral experiments were performed in a 16 T × 16 W × 32 L inch light attenuating plexiglass compartment and were recorded using a camera mounted on the ceiling above the apparatus connected to a Windows 7 PC running Pinnacle Studio 16 (Corel, Menlo Park, CA, USA). A total of 8 rats per group were placed in the center of the chamber and visually inspected and subjectively scored for gait and tail stiffness. Each rat was scored twice by different raters blinded to the rat’s experimental condition to reduce bias. The two scores were averaged. Tail stiffness was scored 1–5 (1—no stiffness; 2—minor stiffness with no tail elevation with ambulation; 3—minor stiffness with elevation during ambulation; 4—moderate stiffness with elevation at rest; 5—severe stiffness with elevation at all times). Gait was scored 1–5 (1—normal movement pattern with no hunching at rest or during ambulation; 2—normal movement patterns with mild hunching at rest but not during ambulation; 3—abnormal movement patterns with mild hunching at rest but not during ambulation; 4—abnormal movement patterns with moderate to severe hunching at rest but not during ambulation; 5—abnormal movement patterns with severe hunching at rest and during ambulation). Open-field crosses were defined as the number of times the animal crossed through the middle 1/3 of the chamber and rears were defined as independent instances of both front paws leaving the ground and being elevated above resting head level.

### 4.7. Elevated-Plus Maze and Chronic Intermittent Ethanol

Chronic intermittent ethanol exposure was produced by random assignment to either air or ethanol groups. The ethanol group underwent 16 h of continuous ethanol exposure followed by 8 h off each day for four days. This was followed by 3 days of abstinence. Testing occurred 24 h after three exposure cycles. Ethanol vapor was delivered to the ethanol inhalation chamber by volatilizing 190 proof ethanol and mixing the ethanol vapor with fresh air at a rate of 10 L/min. The concentration of ethanol in the chamber was monitored at the completion of each cycle. Intoxication was verified visually by noting loss of consciousness following initial injections with increasing tolerance. The elevated-plus maze consisted of two open arms (22 × 4 inches) and two closed arms (22 × 4 with 6 inch walls). A total of 24 animals were placed in the center of the apparatus for 5 min. Time spent in open versus closed arms was calculated and used a measure of anxiety.

### 4.8. Data Collection and Statistical Analysis

For single-unit electrophysiology studies, discriminated spikes were processed with a spike analyzer, digitized with National Instruments hardware and analyzed with National Instruments LabVIEW and IGOR Pro software (Wavemetrics, Lake Oswego, OR, USA). Extracellularly recorded single-unit action potentials were discriminated by a peak detector digital processing LabVIEW algorithm and recorded in 10 s intervals. Firing rate data was averaged across neurons at 10 s intervals and subsequently binned in 50 s intervals for comparisons across time points as previously reported [14]. Experimental groups were averaged across bins and then compared using a one-way ANOVA then corresponding bins were compared with a Student’s *t*-test. Average depression or excitation in firing was calculated from 10–40 min post injection. Baseline firing rate was calculated from the average of the final 60 s of firing rate data before vibrational stimulus and after 5 min of recording to ensure neuron stability. The results from all MStim and control groups were derived from calculations performed on ratemeter records and expressed as means ± SEM. For microdialysis, the area under the curve for the DA peak was extracted and a two-point calibration was used to approximate the DA concentration. All collections were normalized to the final baseline collection before injection occurred. Dopamine release for each time point was expressed as a percentage of baseline ± SEM. They were compared using a one-way ANOVA after which the groups were compared using a Tukey’s posthoc analysis. For behavioral experiments, after blinded scoring, results were compared using a one-way ANOVA after which groups were compared using a Tukey’s posthoc analysis. For USV’s, calls were separated into 15 min bins and the average principal frequencies for the control and treatment groups were compared using a one-way ANOVA.

## Figures and Tables

**Figure 1 ijms-23-12630-f001:**
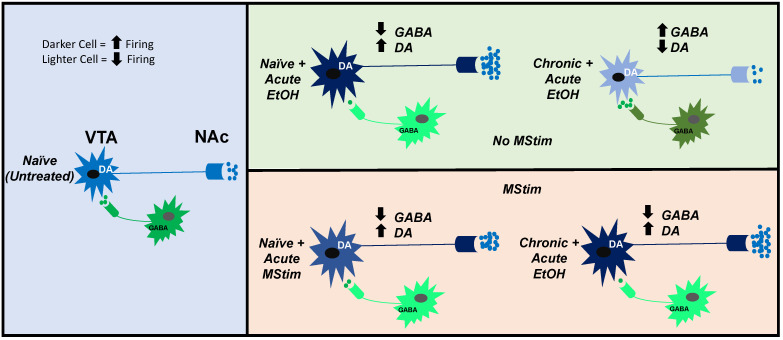
Proposed effects of MStim on VTA GABA and DA neuron firing rate and neurotransmitter release in the VTA and NAc compared between naïve state and acute EtOH exposure in the context of MStim and no MStim and chronic EtOH exposure and no chronic EtOH exposure.

**Figure 2 ijms-23-12630-f002:**
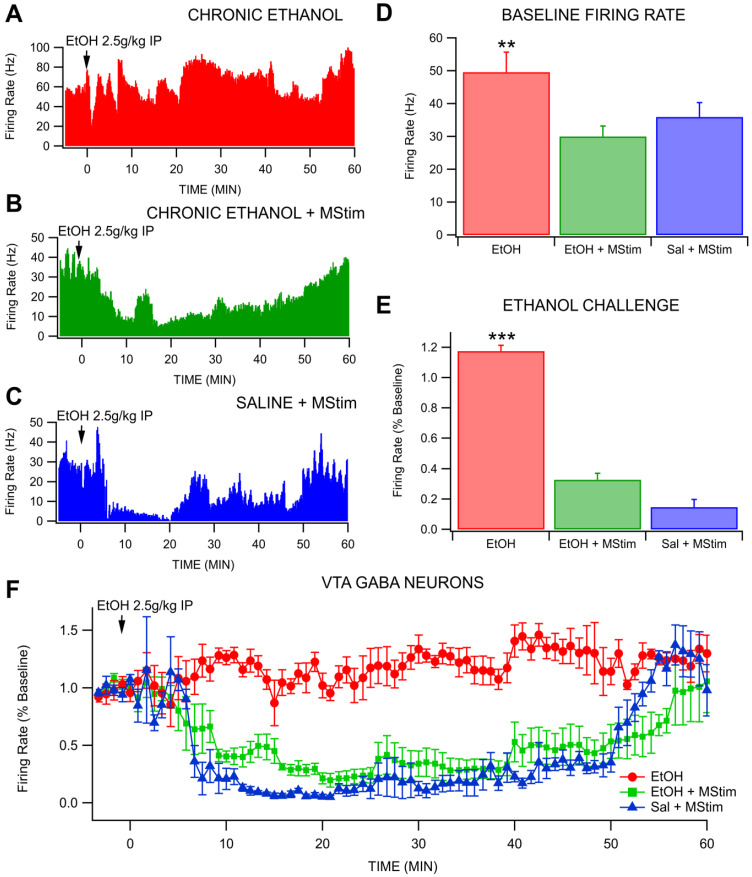
Effects of MStim on VTA GABA neuron firing rate after reinstatement ethanol during withdrawal. (**A**–**C**) Representative traces for GABA neuron response for (**A**) EtOH alone, (**B**) EtOH + MStim and (**C**) saline + MStim. (**D**) Baseline firing rate differences between the three groups. Note that EtOH alone maintained a higher baseline firing rate. (**E**) MStim blocks chronic EtOH-induced desensitization of GABA neurons to EtOH reinstatement. (**F**) Time course data with 50 s bins demonstrating disparate effects among groups. Asterisks (**, ***) indicate significance levels *p* < 0.01 and 0.001, respectively.

**Figure 3 ijms-23-12630-f003:**
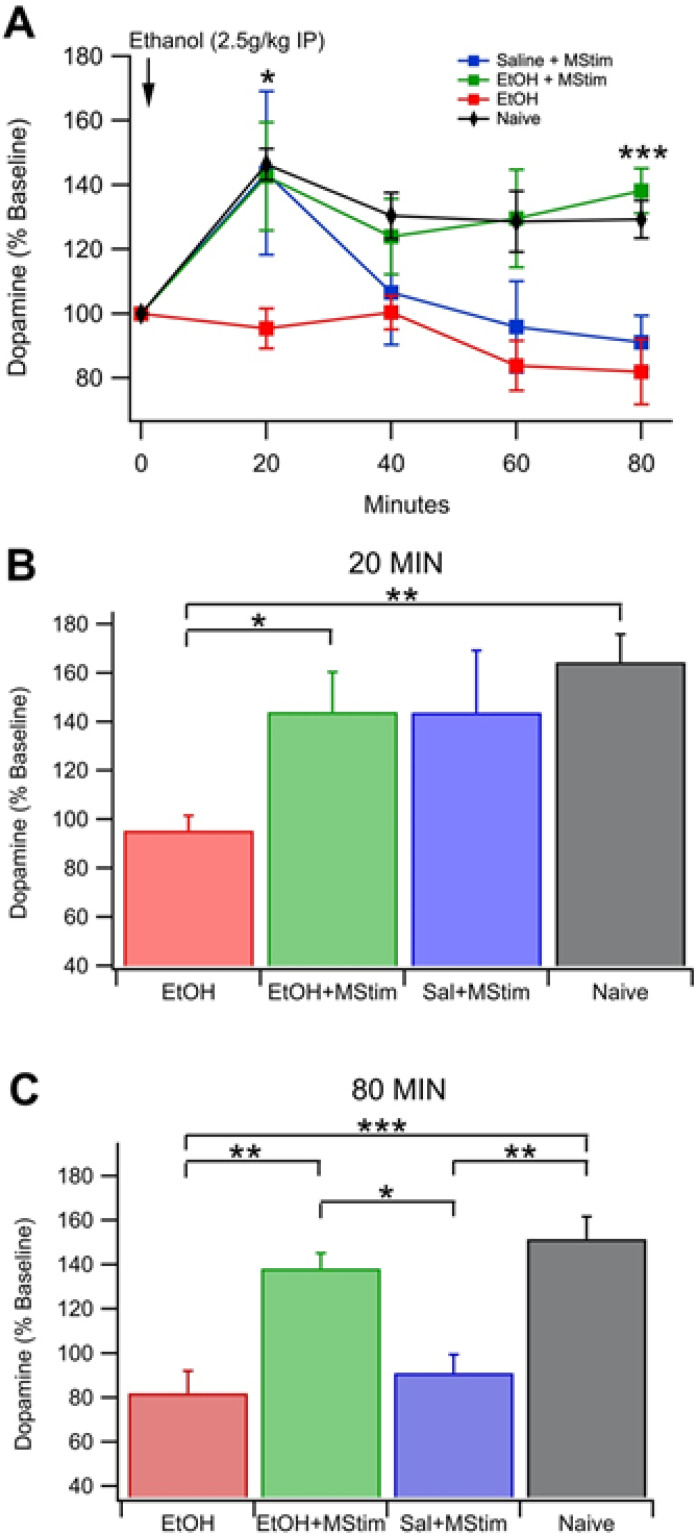
Basal dopamine release in the NAc following EtOH injection (2.5 g/kg IP). (**A**) Summarized time course of EtOH enhancement of basal DA release in the NAc. Note that naïve animals show distinct differences when compared to animal that received EtOH or MStim. (**B**) Comparison of [DA] at 100 min post-injection. (**C**) Comparison of [DA] at 120 min post-injection. Asterisks *, **, *** indicate significance levels *p* < 0.05, 0.01 and 0.001, respectively.

**Figure 4 ijms-23-12630-f004:**
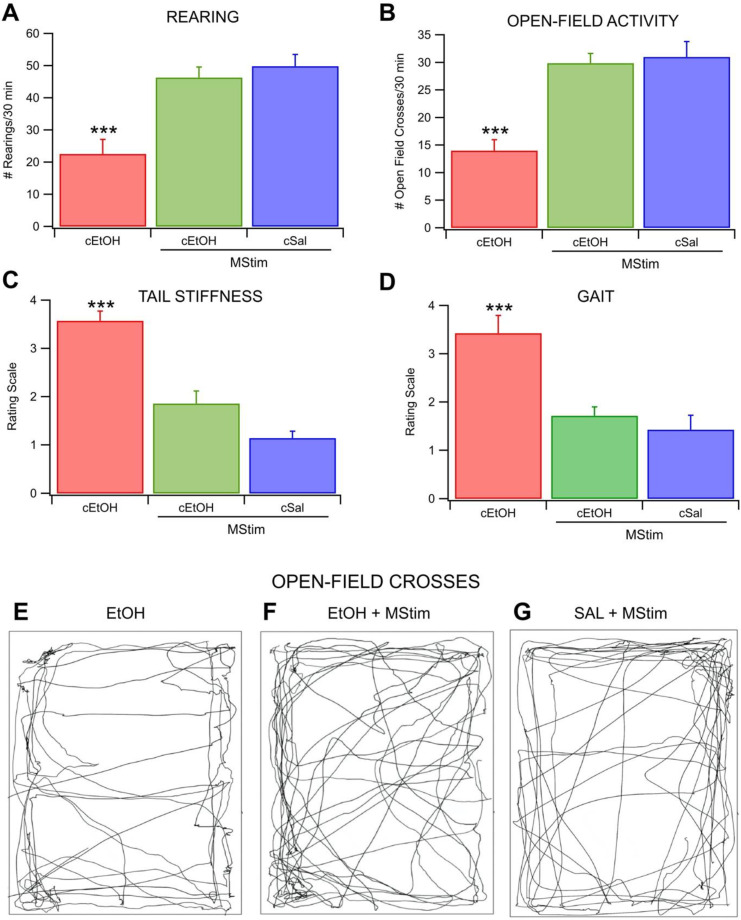
Blocking of EtOH-induced behavioral measures of withdrawal by MStim. (**A**) Number of times the animal reared-up on hind legs in 30 min period. (**B**) Number of times the animal crossed through the middle third of the chamber in a 30 min period. (**C**) Subjective rating of tail-stiffness. (**D**) Subjective measure of the animal’s gait. Note that all measures improved with the concurrent administration of MStim during chronic EtOH exposure. (**E**–**G**) Representative traces of the animals movement patterns during testing. Asterisks (***) indicate significance level *p* < 0.001.

**Figure 5 ijms-23-12630-f005:**
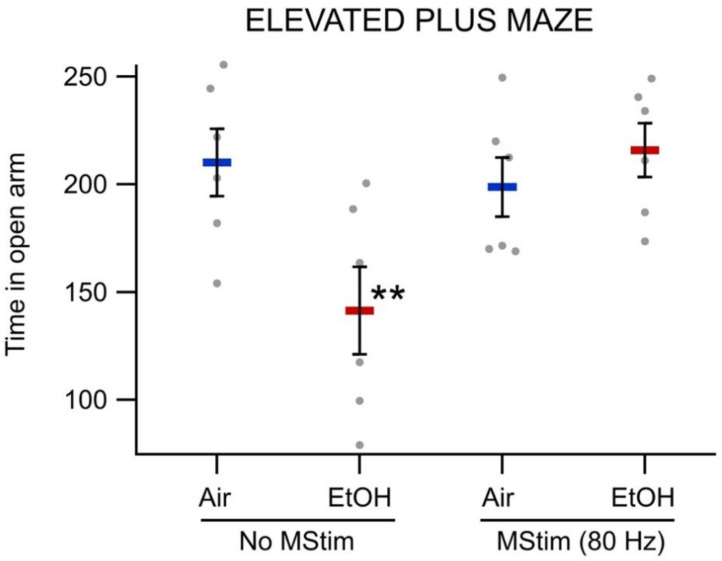
Effects of MStim on Elevated Plus Maze scores. Twenty-four hours following final EtOH or air exposure, animals were placed in Elevated Plus Maze assay and the time spent in the open arm was recorded. Mechanical stimulation significantly increased the time spent in the open arm when compared to EtOH alone. Asterisks (**) indicate significance level *p* < 0.01.

**Figure 6 ijms-23-12630-f006:**
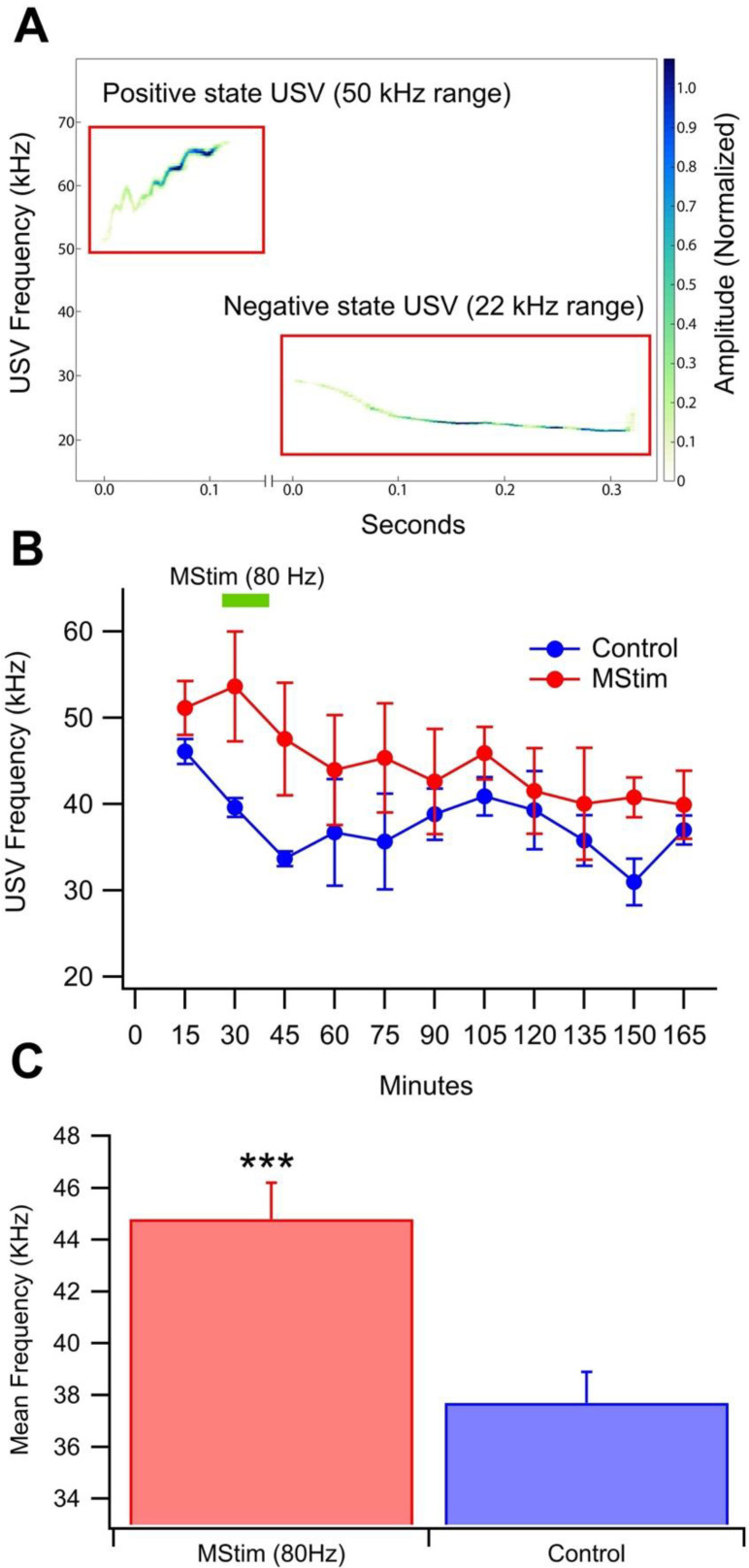
Effect of MStim on the frequency of USVs. USVs were recorded in sound-attenuating chambers during a 2.5 h session. (**A**) Representative traces of calls that fall within the 50 kHz and 22 kHz ranges. (**B**) Time response of the USVs comparing positive and negative calls. (**C**) Comparison of the mean frequency when all times are collapsed between control and MStim. Mechanical stimulation significantly increased the mean vocalization frequency when compared to control, suggesting an induction of a relatively higher positive affective state. Asterisks (***) indicate significance level *p* < 0.001.

## Data Availability

The data supporting the findings of this study are available from the corresponding author upon reasonable request.

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
