# Peer review of "Mechanical Stimulation Alters Chronic Ethanol-Induced Changes to VTA GABA Neurons, NAc DA Release and Measures of Withdrawal"

_ijms, 2022, doi:10.3390/ijms232012630_

Round 1
Reviewer 1 Report
This article title “Mechanical stimulation alters chronic ethanol-induced changes to VTA GABA neurons, NAc DA release and measures of withdrawal” by Bills et al., describes use of active 80 Hz activation of mechanoreceptors in the cervical spine of rat, and its further effects on VTA GABA neurons firing, along with NAc DA release measurement.
This is an incredibly interesting study and has a potential in depicting use of mechano-stimulation physiology and its therapeutic valuation. The overall article is interesting, and collection of information reported are useful in emphasizing critical findings, which directly supports authors theory. I have no concerns related to the literature cited and findings mentioned in this article.
I have some minor concerns, which are described below. Addressing these issues should serve to strengthen this manuscript and increase confidence of readers in the conclusions.
Minor Concerns
1. Line 57: I would recommend author to explain a bit more how 80 Hz stimulation works and why particular 80 Hz stimulation was chosen for mechanostimulation protocol.
2. Line 82: Write hypothesis in assertive sentence.
3. In method section major concern is; it is unclear why only male mice were used as experimental subjects or and why experimental subjects were restricted to a single sex. It is clear that only males were used in this study, however it needs to be explicitly stated in the methods section and a justification needs to be provided for why only male subjects were chosen. Author may choose to provide a section in discussion if both male and female mice were used (hypothetically), and if there were significant effects of sex how do the authors interpret this?
4. Line 107: micropipette used in this study were made of glass borosilicate or quartz?
5. Section 2.2: Provide data regarding how many cells/rat were recorded?
6. Section 2.5: did author balance pH and osmolality of ACSF? If yes, then what solution was used? If not then, what was the actual pH and osmolality of used ACSF?
7. Fig1: Though the firing rate plot convince the findings; It would be more appealing to plot shift in distribution of spikes before and after treatment (80 Hz) in all treatment groups.
Author Response
Reviewer and Editorial Team,
We are grateful for the thoughtful comments and review of the manuscript and the opportunity to improve the manuscript. We have responded to each of the concerns and have amended the manuscript to reflect your suggested changes. We feel your comments and concerns have enhanced the quality of the work. Please see listed below a response to each of the concerns
Reviewer 1
Minor Concerns
- Line 57: I would recommend the author to explain a bit more how 80 Hz stimulation works and why particular 80 Hz stimulation was chosen for mechanostimulation protocol.
Response: The following has been added as requested:
About Line 57:In the previous report, we demonstrated that the central effects disappear with superficial cutaneous stimulation or with higher frequency stimulation
And...
About Line 347: Because previous studies demonstrated that the effects are frequency and anatomically specific, an 80 Hz stimulation was selected and administered as described for this study.
- Line 82: Write hypothesis in assertive sentence.
Response: The following has been added as requested:
We hypothesized that MStim would block chronic EtOH-induced desensitization of VTA GABA neurons and changes in NAc DA release in response to acute EtOH reinstatement and anxiety-related behaviors normally associated with chronic ethanol withdrawal.
- In method section major concern is; it is unclear why only male mice were used as experimental subjects or and why experimental subjects were restricted to a single sex. It is clear that only males were used in this study; however, it needs to be explicitly stated in the methods section and a justification needs to be provided for why only male subjects were chosen. Author may choose to provide a section in discussion if both male and female mice were used (hypothetically), and if there were significant effects of sex how do the authors interpret this?
Response: The following has been added as requested:
About Line 87: Initial MStim testing with male and female rats demonstrated no significant difference in MStim-induced effects on mesolimbic neurons.
- Line 107: micropipette used in this study were made of glass borosilicate or quartz?
Response: The following has been added as requested:
About Line 108: glass borosilicate (BF150-86-10, Sutter Instruments, Novato, CA)
- Section 2.2: Provide data regarding how many cells/rat were recorded?
Response: The following has been added as requested:
About Line 120: One cell was recorded and reported per rat to avoid confounds due to multiple doses of ethanol.
- Section 2.5: did author balance pH and osmolality of ACSF? If yes, then what solution was used? If not then, what was the actual pH and osmolality of used ACSF?
Response: The following has been added as requested:
About Line 153: pH ~7.4 and osmolarity of 300-310 mOSm.
- Fig1: Though the firing rate plot convince the findings; It would be more appealing to plot shift in distribution of spikes before and after treatment (80 Hz) in all treatment groups.
Response: We believe that the rate and distribution of the spikes as a function of frequency and time, before during and after treatment, are visualized in panel A, B, and C of Figure 1.
Reviewer 2 Report
The manuscript is well executed research work by the authors. They clearly showed their data to prove their hypothesis. I would recommend to include a graphical abstract or summary figure to clearly demonstrate the research findings.
Author Response
Reviewer and Editorial Team,
We are grateful for the thoughtful comments and review of the manuscript and the opportunity to improve the manuscript. We have responded to each of the concerns and have amended the manuscript to reflect your suggested changes. We feel your comments and concerns have enhanced the quality of the work. Please see listed below a response to each of the concerns
Reviewer 2
- I would recommend including a graphical abstract or summary figure to clearly demonstrate the research findings.
Response: The requested summary graphic has been added to increase the clarity of findings. Please see attached pdf for image of figure.
